# The Effect of the Spargers Design on the Wastewater Treatment of Gas-Liquid Dispersion Process in a Stirred Tank

**DOI:** 10.3390/e24030357

**Published:** 2022-03-01

**Authors:** Zhi Zheng, Peng Wang, Daolin Du, Qiaorui Si

**Affiliations:** 1Research Center of Fluid Machinery Engineering and Technology, Jiangsu University, Zhenjiang 212013, China; 15985114938@163.com (Z.Z.); siq@163.com (Q.S.); 2School of the Environment and Safety Engineering, Jiangsu University, Zhenjiang 212013, China; daolindu@163.com; 3Hefei Huasheng Pumps & Valves Co., Ltd., No. 006, Jinhui Road, Hefei 231131, China

**Keywords:** wastewater treatment, gas-liquid mixing, power consumption, flow patterns, cavity formation

## Abstract

The hydrodynamic and power characteristics of curved Rushton impeller in an air-wastewater system were investigated using the CFD-PBM method. Studies were conducted primarily in a mixing vessel of diameter 0.39 m. The inference of operating conditions, sparger distribution, and numbers on bulk flow patterns, gas-filled cavity formation, and power consumption have been investigated in detail. It found that the gassed power consumption is closely related to cavity shape and flow patterns. In particular, the development of large cavities causes a significant reduction in power drawn, impeller pumping capacity, and gas dispersion capability. The sparger distribution and location have a strong influence on relative power drawn, power required to disperse gas, and stability of operation. Of the sparger configurations studied, the use of three sparger distributions is suggested, since relative power drawn, gas dispersion capability, and flow patterns in dispersing gas are all enhanced.

## 1. Introduction

The gas-liquid mixture in stirred tank reactors is the important part of multi-phase reaction engineering as it is widely used in biochemical, food and wastewater treatment industries [1,2]. In wastewater treatment process, oxygen is an essential nutrient used by micro-organisms for their growth and metabolism. Gas-liquid stirred tanks are required to reach a wider range of process objectives, such as fermentation, dispersion, oxidation, hydrogenation, and other kinds of processes [3,4]. The effective achievement of a gas-liquid stirred tank affects the degree of gas-liquid homogenization, mixing quality, power consumption, and mass transfer, which will determine the reactions involved in relevant processes [5]. The solubility of air in wastewater is low, making oxygen transfer rate an important entity. Therefore, the mixed product quality and energy consumption are highly related to the flow behavior during the mixing process.

Low gas dispersion rate in stagnant zones, a mark of poor mixing, engender low inter-facial area between the phases, resulting in poor and no-uniform distribution of air in wastewater treatment stirred tanks. The gas-liquid dispersions in stirred tank reactors have been studied extensively, especially the characteristics of the flow field in the mixing tank [6,7,8]. Many parameters have been varied and the performance variables like impeller diameters, superficial gas velocity, gas sparger positions, and sparger diameters have been investigated. Nienow et al. [9] found that using large diameter ring spargers located *T*/25 (where *T* is the tank diameter) below the Rushton turbine blades lead to an increased gas handling capacity, a higher power draw, and as a consequence a higher specific mass transfer coefficient. Joshi et al. [10] reviewed the work on gas flow rate, impeller diameters mechanically stirred gas-liquid tank reactors, and they found that the drag experienced by the bubble depends on both the Reynolds and Eotvos numbers. Hassan and Robinson [11] investigated the effect of sparger separation on flooding case. They found less impact on the disc turbine than the pitched blade down flow turbine impellers. They also found the replacement of the power number versus impeller speed curves with the changes in distance between the sparger and the impeller. Bruijin et al. [12] used pipe and ring spargers of different diameters in the gas-liquid mixing system and observed instabilities in the power consumption when the impeller and the ring sparger were the same diameter. They also found that the increase of the cavities behind the blade leads to the decrease of the gas to ungassed power consumption ratio. Botton et al. [13] also found the importance of the sparger hole diameter and the number of holes for reaching the same superficial velocity for a given flow rate in stirred tanks of different sizes. Birch and Ahmed [14] found that the sparger position is larger than the impeller in the discharge flow from the impeller, which can effectively have avoided. Sardeing et al. [15] reported that the size of sparger did not have significant effect on the mass transfer, whereas the position of the sparger has significant effect. Ohmori et al. [16] presented that a large sparger is more effective for increasing the gas hold-up and mass transfer coefficient. Kamei et al. [17,18] reported the effect of the sparger geometry on power consumption and mass transfer in a gas-liquid agitated vessel using a disk Rushton turbine. They found that a sparger larger than the impeller is effective for obtaining high aeration power consumption and high mass transfer coefficient. In addition, the flooding phenomenon did not occur easily, and the gas is sufficiently dispersed.

Studies on overall gas volume fraction and gas dispersion have been conducted both experimentally and using CFD simulations. Hence, several experimental studies have been undertaken to investigate the flow field of the mixing stirred tank. Rewatkar et al. [19,20] used the complicated laser Doppler anemometer (LDA) to reach the research objective on visual observation of flow patterns of three types of impellers (disc turbine, pitched blade turbine up-flow, and down-flow turbine). They found that the power number have a strong dependence on the flow pattern generated by the impeller. Lee et al. [21] studied the flow patterns using major bubble trajectories in a standard baffled gas-liquid stirred tank with Rushton impeller. They found that the increasing of the rotation speed could improve the distribution of the flow field and the mixing efficiency. To achieve a total homogenization normally requires high energy consumption. Actually, the gas-liquid mixing in the stirred tanks is a highly complex turbulent process, which presents spatiotemporal chaotic behavior [22]. In order to obtain the chaotic behavior characteristics and hydrodynamics with the stirred tanks, computational fluid dynamics (CFD), which involves numerical solution of the Navier Stokes flow equations is widely recognized as an approach to model the hydrodynamics in stirred tanks. Forte et al. [23] utilized CFD and the electrical resistance tomography (ERT) probe to obtain the gas hold-up in a gas-liquid stirred system with a Rushton turbine. They found that the flow regime varied with both impeller speed and gas flow rate; the results were compared with CFD simulation available in the literature, showing good agreement.

Traditionally, disc Rushton turbine impeller (RT) is the most common impeller used in the process of gas-liquid stirred tank. However, many drawbacks of the RT impeller have been point out in this process, such as low-pressure trailing vortices being observed behind the impeller blades and limited gas handing capacity. The RT impeller can only effectively disperse air in the impeller region, and the gas dispersibility was poor in the region far away from the impeller [24,25,26,27,28]. To improve the gas dispersion ability, a lot of high-efficiency impellers were proposed in literatures [29,30,31]. They found that pitched Rushton impeller can enhance the gas dispersion rate and reduce the trailing vortices more than RT impeller in both single and double impeller cases using CFD prediction. Rielly and Nagy [32] utilized the CFD to study the gas-liquid stirred system and pointed out that the concave shape of the RT impeller produced a smaller gas cavity behind the blades.

In summary, it can be found that previous work placed the emphasis on the understanding of the process of gas dispersion and optimization of the impeller design, so as to avoid flooding (flooding is one of the phenomena in gas-liquid mixing when high gas loadings make it so that the impeller no longer pumps the gas and liquid mixture adequately and the gas rises axially as a bubble stream to the liquid surface). Little attention was paid to the systematic optimization of sparger design, sparger location away from impeller and rheological properties of wastewater (it presents non-Newtonian fluids properties) within in gas-liquid stirred tanks. Many researchers used the RT impeller instead of the concave shape impeller in gas-liquid system. Therefore, in this work, a type of concave impeller was proposed based on the RT impeller to improve the gas-liquid dispersion performance. The effects of spargers design and location with a concave impeller on flow patterns and power consumption in non-Newtonian fluids were investigated. The remaining part of the paper is organized as follows: Section 2 provides a detailed description of the experimental set-up and methods for measuring the rheology of non-Newtonian fluids, and of how the torque when operating in different rotation speed characterization was obtained. The CFD methods and basic theory used in this paper are also presented; Section 3.1 firstly provides a brief comparison between the measured torque and the CFD predict value for validating the proposed CFD method, Section 3.2 provides the measured power in different operation conditions and from Section 3.3 to the end, the CFD simulation results are presented. The salient conclusions are highlighted in Section 4.

## 2. Materials and Methods

### 2.1. Experimental Set-Up

The schematic of the experimental facilities for two-phase mixture is shown in Figure 1. The experiments were conducted in a flat-bottom transparent cylindrical vessel of diameter *T* = 0.39 m (details of geometry parameters can be seen in the Table 1). The wastewater demonstrates the non-Newtonian properties, therefore, the transplant carboxymethylcellulose (CMC) solution was used as the model fluid to imitate opaque wastewater. The gas phase was air, and the liquid phase was CMC solution. A sparger system was located at the bottom of the tank, as shown in Figure 1. A six curved blade impeller was employed, as shown in Figure 1b. Since high viscous fluids were used in this study, the viscous effects damped the fluid motion induced by the turbine at low rotational speeds. Therefore, no baffles were used to avoid stagnation zones in front of and behind the baffles. To eliminate the effect of disc on the gas bubbles flow, the impeller was fixed at a clearance off the bottom at 2*T*/3 of the tank diameter. Rotational speed ranged from 10 to 180 rpm. The national instruments data acquisition system establishes the connection between the devices and the host computer for system control and data recording. The power was supplied to the impeller with an electric motor using the air-bearing system to minimize the friction inference on the measurement process. The torque sensor (ETH-Messtechnik GmbH, Gschwend, Germany), mounted on the impeller shaft, was used to measure the power consumption and the impeller corresponding speed. The applied sensor could measure torque values up to 0.5 N·m with an accuracy of 0.1%.

To avoid agglomeration, CMC solution was prepared using distilled water in an unbaffled and jacketed mixing tank stirred with a hand-hold mixer. Specific mass contents of CMC powder were added slowly in the region of the vortex formed around the impeller shaft to permit the particles to become individually wetted and can be incorporated quickly into the mixing solution. Right after a determined amount of CMC powder was added into the system, the solution was stirred for 3 h. Finally, the solution remained at rest for 24 h to allow the releasing of trapped air bubbles. The rheological properties were measured using Brookfied rheometer, and its rheological properties were fitted to a power-law model [33]:(1)$$\mu_{a}=K{\left(\dot{\gamma }\right)}^{m-1}$$
where $\mu_{a}$ is the apparent viscosity, *K* and *m* are the consistency and flow behavior index, respectively. The fitted parameters in the shear rate range of 0.067 s^−1^ < $\dot{\gamma }$ < 860 s^−1^ were *K*_1_ = 1.75, *K*_2_ = 14.99, *K*_3_ = 115.34 Pa·s^m^, *m*_1_ = 0.464, *m*_2_ = 0.388, *m*_3_ = 0.325, which yielded *R*^2^_1,2,3_ = 0.994, 0.996, 0.995, respectively; the regression process that obtained the *K* and *m* are seen in the Table 2 and Figure 2.

### 2.2. Experimental Method

During the experiments under gassed conditions, compressed air was sparged through five orifices in the tank bottom. Three arrangements of air sparging were investigated as shown in Figure 3. Air was introduced via a tube equipped with a certain number (1,3 and 5) of 2.0 mm orifices and the opening of orifices were controlled by an air distributor. The four orifices were evenly distributed on the circumference a diameter of *D*/3 (as seen in Figure 3). Gas flow rates were set as 5, 10, and 15 L/min. The flow meter was controlled by a Labview program, which regulates the volumetric flow rate of the gas. For the measurement, the flow rate was kept at a constant value, and the rotational speed of the turbine was varying from the maximum to minimum.

In the gas-liquid tank, the power consumption was determined by measuring the torque, which was measured by the torque sensor mounted on the stirring shaft. The power consumption was calculated by the following equation:(2)$$P=2\pi N\left(M-M_{0}\right)$$
where *P* is the net power consumption, *M* is measured torque under normal operating condition, and $M_{0}$ is the unloaded torque. Torque sampling frequency was set as 1000 Hz. *P* is the power without sparged gas, and the $P_{0}$ is the gassed power under aeration.

The dimensionless power number (or Newton number) $Ne_{g}$ and Reynolds number *Re*, which could affect the mixing process, were defined as follows:(3)$$Re=\frac{\rho ND^{2}}{\mu_{a}}$$
(4)$$Ne_{g}=\frac{P}{\rho N^{3}D^{5}}$$
where $\mu_{a}$ is the non-Newtonian apparent viscosity, ρ is the fluid density, *N* is the rotational speed, *D* is the agitator diameter and *P* is the power.

### 2.3. CFD Methodology

#### 2.3.1. 3D Geometry and Mesh Generation

The commercial CFD software ANSYS Fluent 19.1 was applied to model the hydrodynamics of single-phase aqueous CMC solutions and gas-CMC solution two-phase mixture in mixing tanks stirred by a curved-blade Rushton turbine. The tank geometry for the two-phase flow were modeled directly in the meshing software, and the sparger system configuration of the gas-liquid mixture, five small cylinders of 15 mm length and 2 mm diameter were modeled at the tank bottom. The 3D calculation domains were imported in ICEM CFD for meshing and pre-processing. Hexahedral structured grids of all parts were generated, in order to have higher accuracy of the numerical calculations. The structured mesh of the whole impeller after mesh rotation of the mesh is shown in Figure 4. The sufficient amount of mesh nodes was created in the rotor zone including the gas sparger inlet to have a much-fined mesh. To verify the mesh independence, the number of cells was increased from the original 345,100 to about 485,000, 676,200, and 956,970 cells, respectively, as seen in Figure 5. These increases changed the velocity and power number in rotor region by less than 3%, however, the calculation time were around 20, 30, 50 h, respectively. Finally, to save the calculation resource, the number of 676,200 cells were employed to numerical simulations.

#### 2.3.2. Numerical Simulation of Gas-Liquid Two-Phase Mixing System

The Eulerian–Eulerian multiphase approach was implemented in this study to simulate the 3D geometry of the experimental rig. The sliding mesh method has been applied in this work to simulate the unsteady state behavior of multiphase system hydrodynamics. The total interfacial force acting between the two phases may arise from several independent physical effects, such as the forces caused by interfacial drag, lift, mass, turbulence dispersion, and pressure gradient. The non-Newtonian fluid rheology properties was modeled with the power law model as demonstrated in Equation (1).
(5)$$\frac{\partial }{\partial t}\left(\alpha_{i}\rho_{i}\right)+\nabla \cdot \left(\alpha_{i}\rho_{i}v_{i}\right)=0$$
where $\alpha_{i}$ is the volume fraction of the continuous phase, and $v_{i}$ is the liquid mean velocity.
(6)$$\frac{\partial }{\partial t}\left(\alpha_{i}\rho_{i}v_{i}\right)+\nabla \cdot \left(\alpha_{i}\rho_{i}v_{i}⊗v_{i}\right)=-\alpha_{i}\nabla p^{\prime }+g\alpha_{i}\rho_{i}+\nabla \cdot \left[\alpha_{i}\mu_{a,i}\left(\nabla v_{i}+\nabla v_{i}{}^{T}\right)\right]-\frac{2}{3}\nabla \left(\alpha_{i}\mu_{a,i}\nabla v_{i}\right)+\sum_{i=1}^{n}\vec{F_{i}}$$
(7)$$\sum_{i=1,2}\alpha_{i}=1$$
where $\mu_{a,i}$ indicates the liquid effective viscosity, $\overset{n}{\underset{i\;=\;1}{\sum }}\vec{F_{i}}$ shows the momentum transferred from bubbles to the liquid phase and the last term in the right side of Equation (6) is the inter-phase forces.

The standard k_ε turbulence model for liquid and mixture turbulence multi-phase was used. Here, the two phases of the mixture share the same turbulence quantities. The corresponding equations for turbulent kinetic energy *k* and its dissipation rate can be written as:(8)$$\frac{\partial \left(\rho_{m}k\right)}{\partial t}+\nabla \cdot \left(\rho_{m}v_{m}k\right)=\nabla \cdot \left(\frac{\eta_{t}}{\sigma_{k}}\nabla k\right)+G-\rho_{m}\varepsilon$$
(9)$$\frac{\partial \left(\rho_{m}\varepsilon \right)}{\partial t}+\nabla \cdot \left(\rho_{m}v_{m}\varepsilon \right)=\nabla \cdot \left(\frac{\eta_{t}}{\sigma_{\varepsilon }}\nabla \varepsilon \right)+\frac{\varepsilon }{k}\left(C_{1}G-C_{2}\rho_{m}\varepsilon \right)$$
(10)$$G=\eta_{t}\left(\nabla V_{k}+\nabla V_{k}{}^{t}\right):\nabla V_{k}$$
where *G* represents the turbulence generation rate, *k* is the turbulent kinetic energy, ε is the dissipation rate and $\eta_{t}$ is the turbulent viscosity, $\sigma_{k}$ and $\sigma_{\varepsilon }$ are turbulent Prandtl numbers. The mixture properties can be found from the following equations:(11)$$\eta_{t,m}=\rho_{m}C_{\eta }\frac{k^{2}}{\varepsilon }$$
(12)$$\rho_{m}=\sum_{i=1}^{n}\alpha_{i}\rho_{i}$$
(13)$$v_{m}=\frac{\sum_{i=1-1}^{n}\alpha_{i}\rho_{i}\vec{v_{i}}}{\sum_{i=1}^{n}\alpha_{i}\rho_{i}}$$

The standard values of the parameters have been chosen as follows: $C_{\eta }\;$= 0.09, $C_{1}=1$.44, $C_{2}=1$.92, $\sigma_{k}\;$= 1, and $\sigma_{\varepsilon }\;$= 1.3, further, values for these equations were extracted from previous literature [34].

The drag force $F_{D}$ for inter-phase momentum exchange, which dominates the motion of bubbles in the bulk has been reported by Mahmoud et al. (1990) [35] as:(14)$$F_{D}=0.75\rho_{F}\alpha_{G}\frac{C_{D}}{d_{b}}\left|V_{L}-V_{G}\right|\left(V_{L}-V_{G}\right)$$

The parameter $F_{D}$ is the drag coefficient and is calculated by Schiller–Nauman’s drag model as:(15)$$C_{D}=\frac{24}{Re_{b}}\left(1+0.15Re_{b}{}^{0.687}\right),\;\;\;\;Re\leq 1000\;$$
(16)$$C_{D}=0.44,\;\;Re\geq 1000$$

In many previous studies [34,36,37], the gas phase bubble size has often been assumed to be constant for simplicity and reduce the computational resource. However, in a real gas-liquid system, the bubbles come through the spargers and undergo breakage and coalescence due to turbulent flow conditions; this results in the change of diameter of the bubbles. To model the bubble size and distributions, a population balance model (PBM) is used in the present work. The bubble size can change during the mixing process of non-Newtonian fluids. Hence, the CFD-PBM coupled method was used to calculate the bubble size change such as coalescence and breakage [38,39]. The PBM method describes the bubble in/out defined control volume.
(17)$$\frac{\partial \left(\rho_{p},a_{i}\right)}{\partial t}+\nabla \left[\rho_{p},a_{i}u_{i}\right]+\nabla V\left(\frac{G_{v_{i}}\rho_{p},a_{i}}{V}\right)=\rho_{p},V_{i}\left(B_{ag,i}-D_{ag,i}+B_{br,i}-D_{br,i}\right)$$
where the balance equation in terms of volume fraction of particle ‘*i*’ was expressed, $\rho_{p}$ is the density of secondary phase ‘*p*’ (kgm^−^^3^), $a_{i}$ is the volume fraction of particle ’*i*’ (%), $V_{i}$ is the volume of the particle ‘*i*’ (m^3^), $B_{ag,i}$ is the particle birth rate due to aggregation (m^−3^s^−1^), $D_{ag,i}$ is the particle death rate due to aggregation (m^−3^s^−1^), $B_{br,i}$ is the particle birth rate due to breakage (m^−3^s^−1^), $D_{br,i}$ is the particle death rate due to breakage (m^−3^s^−1^). The growth rate of particle ‘*i*’, which is $G_{v_{i}}$(m^3^s^−1^) of volume ‘*V*_i_’, is defined as:(18)$$G_{v_{i}}=\frac{\partial V_{i}}{\partial t}$$

The volume fraction of particle ‘*i*’, *α*_i_ (%) was defined as Equation (18), where $N_{i}$ is the local average number density of particle ‘*i*’ (m^−3^s^−1^) and was expressed as Equation (19), where, n(V,t) is the number density function.
(19)$$a_{i}=\sum_{t}N_{i}V_{i}$$
(20)$$N_{i}=\int_{V_{i}}^{V_{i}+1}n\left(V_{i},t\right)dV$$

Impeller rotation speeds were varied between 10 and 180 rpm to model the flow dynamics in the laminar regime. The velocity inlet boundary condition was set to sparger entry surfaces at tank bottom. Gas was supplied at three different rates: 5, 10, and 15 L/min corresponding to velocity magnitudes $V_{inj}$ of 4.24, 8.48 and 12.73 m/s, respectively. The gas injection velocity $V_{inj}$ at the inlet of diameter $d_{inj}$ was calculated as follows:(21)$$V_{inj}=4\frac{Q_{G}}{\pi d_{inl}^{2}}$$

It is important to notice that these rate values are related to one-inlet sparger. For 3-inlets and 5-inlets sparger systems, the above values are equally distributed to the different inlets. For example, a 4.24 m/s velocity magnitude at 1 inlet would have a value of 1.41 m/s and 0.85 m/s at each inlet for 3 and 5-inlets sparger, respectively. Turbulent hydraulic diameter setting was preferred to turbulence intensity in this work for better accuracy. No-slip boundary conditions were applied at the tank walls and appear in calculations in form of standard wall functions. Pressure outlet condition was used at the liquid free surface. A gas volume fraction of 1 was set to sparger inlet and surface pressure outlet. The QUICK (Quadratic Upwind Interpolation for Convective Kinetics) discretization scheme was used for momentum, volume of fraction, *k* and ε, since it has been reported to be suited to structured meshes, due to its higher accuracy. Convergence criteria of 10^−5^ was used for all residuals. For transient calculation, the time step size has been defined following the model of Wu (2010), who estimated time step size as the ratio of mesh cell size to gas inlet velocity. He recommended the following correlation for estimating the mesh cell size:(22)$$\frac{\Delta y}{y^{+}}\approx 5.06TRe^{\frac{-7}{8}}$$

The term Δy represents the distance between the wall and center of the first mesh cell, and $y^{+}$ takes ideally the value 1, and *T* is the diameter of the mixing tank. In order to target at least nine impeller revolutions, the number of time steps was set to 1000 with 50 iterations per time step. The time step size was increased with decreasing impeller speed.

## 3. Results and Discussion

### 3.1. Experimental Validation of CFD Model

To enhance accuracy in the results presented in the investigation, the validation of power consumption has been conducted under different sampling frequency. The uncertain analysis of repeatability for devices test has been investigated by the relative uncertainty, which was defined as the ratio of standard deviation to mean value. Figure 6 shows the variation of power consumption of gas flow rates, inlet distributions and fluid concentration at different sampling frequency. The logarithmic plots of the gas-liquid power number $Ne_{g}$ vs. the Reynolds number *Re* are used to compare with the experimental data and CFD predictions. Figure 6a–c shows three cases of power curves with three fluids concentrations, flow rate and inlet distributions used in this work. Since for each concentration, there has many combinations of inlet distributions and gas flow rates. Hence, the CFD predictions of 0.62%, 5 L/min 1 inlet and 0.85%, 10 L/min 3 inlets as a random sample were present here for discussion. The standard deviation bar for the experimental testing fluids is also shown. The differences between CFD and experiment measurement of the power consumption were within ±3.42%, ±5.10%, respectively. A good agreement of the simulation and experimental results is also observed. However, for 1.25%–15 L/min-5 inlets case, the predicted results were lower than the experimental results in the whole range of *Re*. The reason can be that the higher concentration fluids has higher viscous force, which hold the gas within the mixing tank, but the CFD result did not take this kind of viscous force into account during the simulation. Then the selected sampling frequency can be deemed to have been validated and the results to be repeatable in lower concentration fluids. The uncertainty analysis of the torque and power consumption was carried out with three repeatability experiments. When the sampling frequency is determined, each of torque and power consumption in the experimental system is determined by the average of three repeated measurements under the consideration of the uncertainty analysis of devices test. Therefore, the CFD model developed in this work is suitable for predicting the power draw of the investigated impeller in the gas-liquid mixing of lower concentration shear-thinning fluids in the laminar and the transitional region.

### 3.2. Experimental Observations of Sparger Distribution on Power Consumption

In order to investigate the effect of the sparger distribution on the impeller power draw, the gas-liquid power numbers of different flow rates in different inlet distributions were plotted against the Reynolds number in log-log scale for each case. The experiments were carried out in three mass concentration solutions (0.62%, 0.85%, and 1.25%). The gas-liquid mixture power curves of 0.62% for 1, 3, and 5 spargers distributions are presented in Figure 7a–c, respectively. In this concentration, some differences among the power curves were observed at lower Reynolds numbers. At low impeller speeds, the liquid flow generated by the impeller is weak. The gas bubbles generated at the sparger rise vertically through the impeller without any disturbance. Therefore, an increase of the power number with increasing gas flow rate for different inlet distributions was observed. However, when *Re* > 50, the power curves of three distributions overlapped until the end of the curves. It can be explained that for the shear thinning fluids, when the rotation of impeller reaches a specific speed, the power curves were independent with the gas flow rate and sparger distributions. The power curves of 0.85% for three inlets distributions were presented in Figure 7d–f. It can be clearly observed that the power curves completely independent with three sparger distributions and gas flow rates. The power curves of those three cases nearly overlapped in all *Re* region, except one point in 1-inlet case demonstrates a slight deviation. It can be explained by that with the increase of the fluid concentration, the viscous force can hold the gas bubbles, which is limited within the mixing system. Therefore, no matter which kind of inlet distribution or gas flow rates, the total density of the gas-liquid mixture is constant. It means that the circulation of the liquid is mainly generated by gas sparging. The power curves of 1.25% for three distribution cases were presented in Figure 7g–i. It demonstrated that the power curves of the three cases completely independent with the gas flow rate and sparger distributions, which means the power curves overlapped for three gas flow rates. The total density of the mixture is without significant difference with the change of gas flow rate and sparger distributions, the reason being that the power curves of different cases overlapped. However, the fluid concentration in this case is higher, which demonstrates higher viscous force, therefore, the power consumption is the highest one when compared with that of the 0.62% and 0.85% cases.

### 3.3. Numerical Assessment of the Effect of Sparger Distribution on Mixing Flow Patterns

The hydrodynamics behavior in mixing tanks stirred by radial flow impellers under aerated conditions is more complex and still not fully understood. Just as in one-phase mixing, primary and second flow patterns also occur in multiphase mixing. However, the fluid motion induced by the rotating impeller is affected by the gas flow rate and the spargers’ location. Hence, the characteristics of the flow patterns become strictly dependent on the sparger distribution or the agitator speed when the flow rate is very high, or the impeller rotational speed is very low. Figure 8a represents the gas volume fraction distribution in 0.62% CMC with one sparger variation of gas flow rates when operating at 20 rpm. At the 5 L/min case, the gas flow pattern between the sparger and the impeller generated by the impeller is weak; it can be seen that the gas bubbles generated at the sparger rise vertically through the fluid and hit the disc of the impeller without any disturbance around the impeller region before they get out of the moving region. Above the impeller, the gas bubbles rise vertically from the impeller tip to the mixing tank before exiting from the surface. This is due to that the liquid circulation generated by the impeller has a small effect on the gas phase. When the gas flow rate increases to 10 L/min, the increasing volume of gas begins to represent a small circulation below the impeller; this means that the impeller action is not strong enough to effect a remarkable change in the flow behavior of the gas phase, but the gas bubbles themselves. When the gas flow rate keeps increasing to 15 L/min, the circulation of the high gas volume bubbles below the impeller increased at the same time. As the gas passes through the impeller region, the gas phases occupied the region above the impeller before they left the mixing tank. Since bubble break-up does not occur through the impeller action in this region, the impeller is said to be flooded with gas, and this state is called the flooding.

Figure 8b represents the effect of the sparger numbers on the gas volume fraction distribution of 0.62% CMC when operating at the 180 rpm and 5 L/min cases. In the one sparger case, it can be seen that the flow patterns between the spargers and the impeller are the same as the 20 rpm case. There was no disturbance in this region, and the gas bubbles rose vertically to the impeller surface. The gas bubbles were away from the impeller tip and concentrate as one column before getting out of the mixing tank. When the sparger increases to three, it can be seen that as the total flow rate was shared by three spargers, the high-volume fraction was only present near the sparger region. It can be clearly seen that the gas bubbles from two other spargers rose vertically to the surface of the tank without any action from the impeller. Therefore, only the middle sparger gas bubbles were dispersed by the impeller, and the intensity of eddy motion behind the impeller decreased and the turbulent energy dissipation there was reduced. When the spargers increased to five, it can be seen that the gas bubble column from the middle sparger was even smaller when compared with one and three spargers’ distributions. Due to the total gas flow rate shared by five spargers, the pressure from the spargers decreased, which resulted in that the gas bubbles from the other four spargers dispersed not vertically. In the impeller region, the gas bubbles were distributed in the radial direction above the plane of the impeller as the result of the impeller action. Afterward, the rose to the liquid surface. In these conditions, there is barely or no gas, which recirculates in the area below the impeller disc. This occurrence characterizes the transition from flooding to loading state.

Figure 9a represents the effect of the sparger numbers on the CMC velocity field distribution of 0.62% CMC when operating at 20 rpm and 5 L/min cases. It can be seen that in one sparger case, the high-speed rising of the gas bubbles resulted in two circulations of fluids below the impeller. It means that the flow field below the impeller was mainly dispersed by the gas bubbles. In the impeller region, the high flow field also comes from the gas phase and there were small amounts of radial fluid as a result of impeller action. There were two circulation loops above the impeller region, the two circulations were also a result of the gas bubbles corresponding to the Figure 9a. When the sparger increased to three, the two circulations below the impeller disappeared as a result of the three high speed. The circulations were replaced by two high velocity jets until they reached the impeller, and then kept rising to above the impeller region. In the rotation domain, the moving of CMC, mainly as results of the gas bubbles and the impeller, has no significant action on the fluids. The flow field above the impeller region did not demonstrate any circulations since the gas flow rate was divided into three spargers and the action force and velocity on the fluid obviously weaken when compared that of one sparger. As the sparger kept increasing to five, the high-speed movement of the fluids near the spargers significantly decreased. Due to the decreasing of gas superficial velocity, it can be seen that the impeller action distributes the CMC fluid in the impeller region in the radial direction. Two small scale circulation flows are present at the impeller tips as a result of those radial flows.

The increasing of rotation speed to 180 rpm, the CMC velocity field distribution of 0.62% CMC when operating at 1800 rpm and 5 L/min cases are presented in Figure 9b. It can be seen that the high velocity directly hit the disc of the impeller and was divided into radial direction to the tank wall, which formed two vortexes below and up the impeller blades. The higher vector kept moving to the surface, consequently, another two obvious vortexes presented in the near surface area. With the increasing of sparger to three, the higher velocity from the sparger was replaced by three jets and hit the disc of the impeller. Compared with one sparger case, the radial direction vortex below and up the impeller blades was not obvious, and the upside demonstrates high stream before coming out of the surface. The flow was divided into four parts in the plane below the impeller, which was a result of the high speed of the gas phase from the sparger. When the sparger increased to five, the superficial gas velocity from the sparger decreased, but the flow field near the impeller did not make a significant change, since the rotation speed of the impeller operated at 180 rpm. However, one of the vortexes above the impeller blades disappeared and was replaced by the even flow of CMC fluids when compared with one and three spargers’ cases. It can be explained that since the superficial velocity of gas decreased, high velocity field resulting from the gas flow also decreased. As the flow reached the surface of the tank, there was not enough strength to move the fluid in this area, and the fluid did not mix and no vortex was presented in the right side up the impeller.

Therefore, the characteristics of the flow patterns were affected by the gas flow rate, rotating speed, and the sparger distributions. It can be generally described by three main states as shown in Figure 10. As discussed in Figure 8a, when the impeller operated at lower speed or very high gas flow rates, where the sparged gas rises through the impeller to the fluid surface without being dispersed. In this case, the impeller is said to be in a flooding state. When the gas flow rate decreased or the speed decreased to a specific value as the spargers increased, it leads to a radial discharge of the gas to the mixing tank. It corresponds to what is shown in Figure 9a, where the gas will then rise to the liquid surface. In these conditions, there is barely or no gas, which recirculated in the area below the impeller. This occurrence characterizes the transition from flooding to loading state. When further increasing the impeller rotational speed or decreasing the flow rate, the gas recirculated throughout the whole vessel, and a complete dispersion occurred. The impeller broke up the gas bubbles before they were distributed in the tank. This phenomenon at an impeller speed beyond which no further change of the flow patterns in the vessel was observed, is called complete dispersion.

It was explained by the fact that in low viscous fluids and at low impeller speeds, the axial forces of the gas flow jet dominate the viscous forces and the impeller centrifugal forces. No radial discharge of the impeller toward the tank walls was observed at impeller speed *N* = 20 rpm. This effect is more visible with the 1-inlet sparger, where the whole gas flows through the center hole and hits the impeller disc. Logically, it can be observed that the decrease of those differences from the 3-inlets to the 5-inlets as the input flow rate was distributed equally among the 3- and 5-inlets entries, respectively. At higher impeller speeds (from Reynolds number *Re*_g_ = 60), the effect of the flow rate was damped by the impeller centrifugal forces and the power number is no more dependent on the flow rate. Hence, the power curves overlap. It was illustrated that the radial forces are high enough to overcome the force of the flow jet unlike the impeller action at *N* = 20 rpm. However, in the case of 3-inlets, a slight decrease of the power number with decreased gas flow rate was observed from Reynolds equal to 30.

### 3.4. Numerical Assessment of the Effect of Sparger Distribution on Cavity Formation

Impeller rotation generates a low-pressure region behind the impeller blades. At a specific impeller speed, the regions of low pressure behind the blade promote the gas collection, against the buoyancy force. The gas accumulated behind the impeller is referred to as a cavity. This occurrence results in a reduction of power draw with the increasing impeller speed. The cavity size and shape are dependent on the flow rate and the impeller speed. Therefore, we can neglect the inference of the flow rate, which was fixed at 5 L/min. In Figure 11a, the sparger distributions of 1,3, and 5 inlets with different rotation speeds and their effect on cavity in 5 L/min of 0.62% CMC fluids are presented. It can be seen that in one sparger case, only small amounts of the gas bubbles accumulated behind the impeller, and no cavity structures can be found near the tank wall region. In three spargers with speed of 100 rpm case, the cavity broke up and appear continuously in t the impeller region. In the five spargers case, when the rotation speed reached 180 rpm, the cavity structures behind the impeller broke up and were unevenly distributed in the tank region. The gas bubbles penetrated into the region below the impeller plane, and even reached the tank bottom. The cavity distribution of 5 L/min of 0.85% CMC is demonstrated in Figure 11b; it can be seen that there small amounts of cavities accumulated behind the impeller in the one sparger 50 rpm case. The most important thing is that the cavity structures are unevenly distributed behind the blades. Besides, the broken cavity structures moved to the near wall region and also presented as unevenly ring distributed. When the rotation speed increased to 100 rpm with three spargers, it can be seen that the cavity structures were dropped from the blades, randomly distributed between the blades. As result of the increasing centrifugal force, the cavity ring outside of the impeller was broken and asymmetrically presented. This phenomenon was more obvious when the rotation speed increased to 180 rpm with five spargers case. With further increase of the spargers, the cavities get bigger and begin to cling at the outer extremity of the blades. Hence, a transition from vortex to clinging cavity structure is observed. In this case, there was some random high concentration of cavities structures behind the impeller. The cavity structures distribution with 5 L/min,1.25% CMC are demonstrated in Figure 11c; it can be seen that there were two cavities developed from blades to the near wall region in the one sparger case. As result of the higher viscous force in this concentration fluid, the gas bubbles mainly hold within the mixing tank in the region outside the rotation domain. When the spargers increased to three with 100 rpm, the gas bubbles were broken by the moving impeller and the cavities were also unevenly distributed within the mixing tank. Compared with one sparger, the increasing spargers helps the gas bubbles distributed more evenly in the investigated plane. Only small-sized cavities were observed behind the blades and dropped from the tip of impeller. However, when the spargers increased to five under 180 rpm, there were even cavities shown behind the blades and the size of the structures were nearly the same. Large cavities developed when the sparger was further increased. The radial flow of the impeller was inhibited, and a poor gas distribution occured. A “3-3 structure” cavity type with both larger and smaller cavities is generally a particularity of flat-blade disc impellers. With high centrifugal force, those cavity structures fell from the blades tip and developed to the stationary area.

## 4. Conclusions

A combination of CFD and PBM was applied to investigate the performance of a concave Rushton impeller on the gas-liquid power reduction characteristics. Several independent factors affecting the gas-liquid power reduction characteristics were analyzed, including fluid concentration, gas flow rates, rotation speed, gas inlet distribution, and gas cavity accumulation behind the blades. The results of CFD simulation were verified by experimental results with lower error.

The effect of the sparger distribution on the power consumption shows that at low impeller speeds, the liquid flow generated by the impeller is weak. The gas bubbles generated at the sparger rise vertically through the impeller without any disturbance. An increase of the power number with increasing gas flow rate for different inlet distributions was observed. With the increase of the fluid concentration, the viscous force can hold the gas bubbles, which is limited within the mixing system. In higher concentration fluids, the viscous force holds the gas bubbles, then no matter what kind of inlet distribution or gas flow rates, the total density of the gas-liquid mixture is constant; thus, the circulation of the liquid is mainly generated by gas sparging. Therefore, in order to obtain a good flow field, a three spargers distribution case is recommended in higher concentration fluid mixing since the power curves of different cases overlapped.

The sparger distribution on gas volume fraction demonstrates that at lower rotation speed and lower gas flow rates, the flow circulations within the mixing tank mainly comes from the gas bubbles from the spargers. With the increasing of gas flow rates, which results in a small circulation below and up the impeller region. It means the impeller action is not strong enough to effect a remarkable change in flow behavior of the gas phase. In this case, the flow is recognized as the flooding state. However, when the rotation speed is 180 rpm, the higher radial flow from the impeller can be observed, but hit the tank wall, especially in three spargers case. In this case, the high superficial gas velocity was divided into three, and as result of the moving fluids, the gas bubbles spiraled up to the surface, and did not disperse by the impeller and the characteristics transition from flooding to loading state. Therefore, the intensity of eddy motion behind the impeller decreases and the turbulent energy dissipation there is reduced. In five spargers case, the gas bubbles are distributed in the radial direction above the plane of the impeller; there is barely or no gas that recirculates the fluids in those areas. Therefore, three spargers are also recommended when referring to the gas bubble circulation of the fluids where the complete dispersion state was observed.

The effect of the sparger distribution on the cavity formation demonstrated that in lower gas flow rate and one sparger case, only small amounts of gas bubbles accumulated behind the impeller, and without cavity structure formed. In three spargers with speed of 100 rpm case, the cavity broke up and continuously appears in the impeller region. The increasing of the rheological concentration resulted in the uneven cavity formation and break up behind the blades. In five spargers, the cavities got bigger and began to cling at the outer extremity of the blades, and a transition from vortex to clinging cavity structures was observed. Increasing of the spargers resulted in a poor gas distribution, and a “3-3 structure” cavity type with both larger and smaller cavities, which is generally a particularity of flat-blade disc impellers.

## Figures and Tables

**Figure 1 entropy-24-00357-f001:**
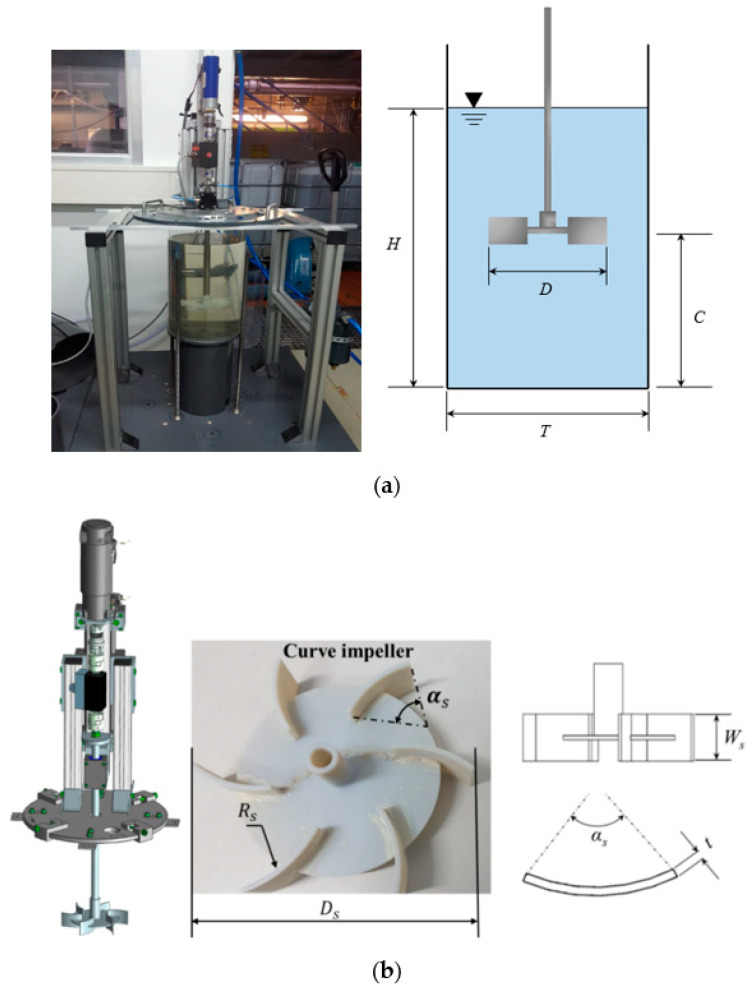
Experimental set-up and the curved-blade Rushton turbine parameters. (**a**) The experimental facilities. (**b**) Parameters of the new curved-blade Rushton turbines.

**Figure 2 entropy-24-00357-f002:**
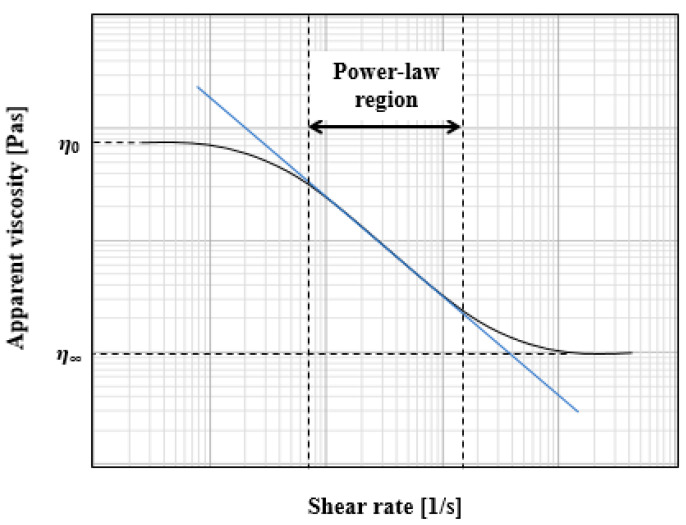
The fitted rheological parameters of power-law fluids, the blue line is the fitting progressivity line.

**Figure 3 entropy-24-00357-f003:**
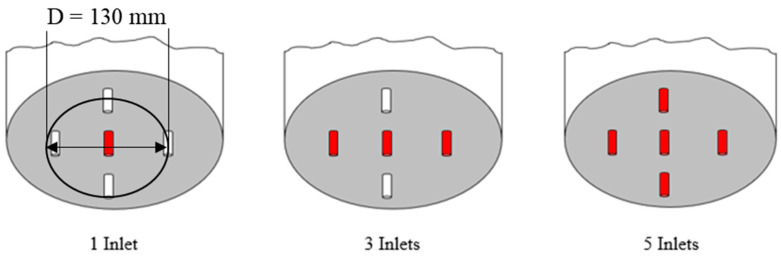
Sparger arrangements for gassed mixing systems.

**Figure 4 entropy-24-00357-f004:**
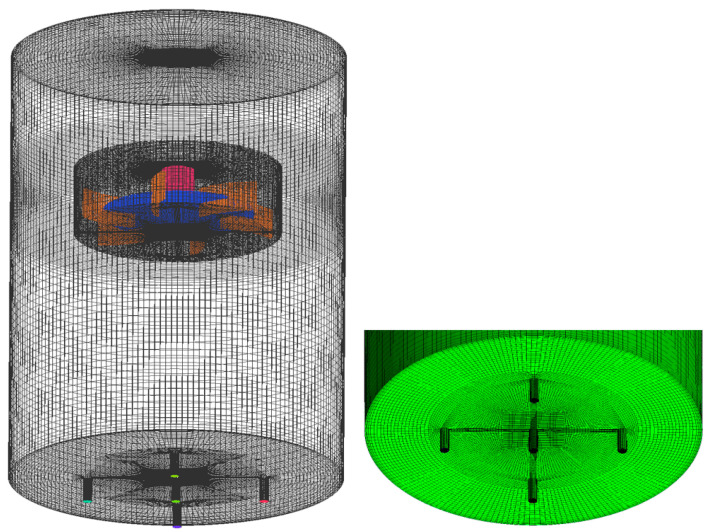
The structured mesh of the computational domains.

**Figure 5 entropy-24-00357-f005:**
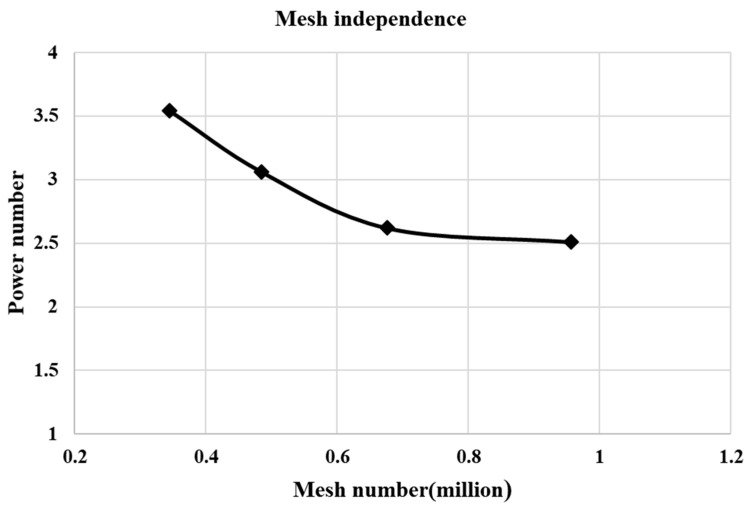
The mesh independence analysis of the power number under 100 rpm at 0.62%.

**Figure 6 entropy-24-00357-f006:**
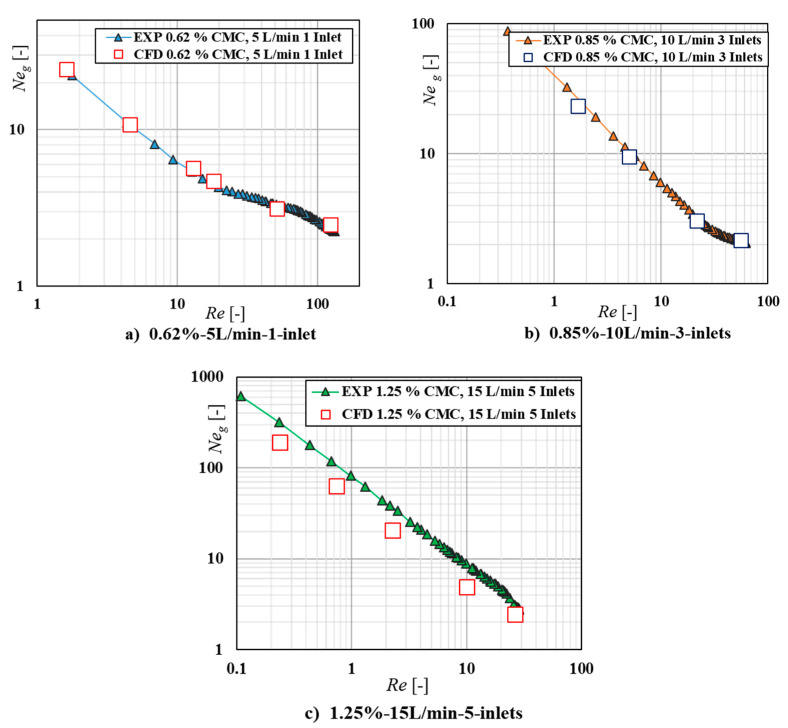
Comparison of power curves between experimental and CFD in different rheology fluids, flow rates, and inlets cases.

**Figure 7 entropy-24-00357-f007:**
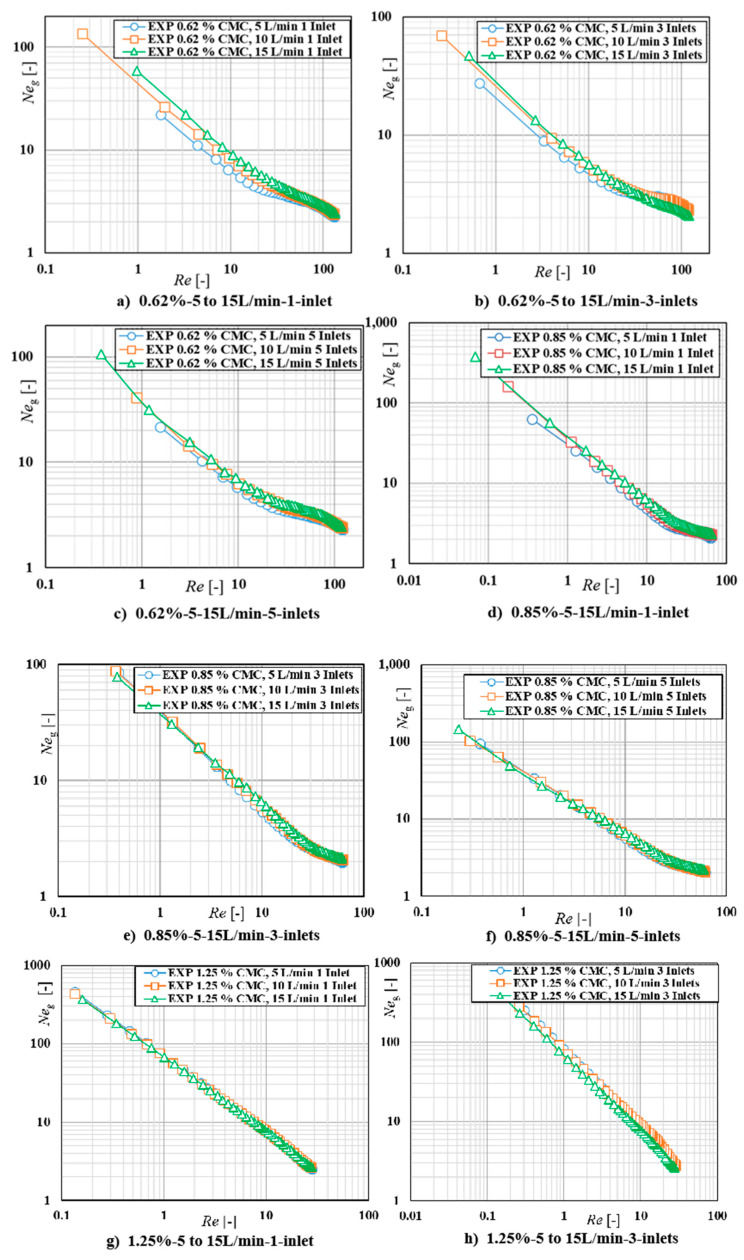
Power curves of different inlet distributions at different flow rates in three concentrations.

**Figure 8 entropy-24-00357-f008:**
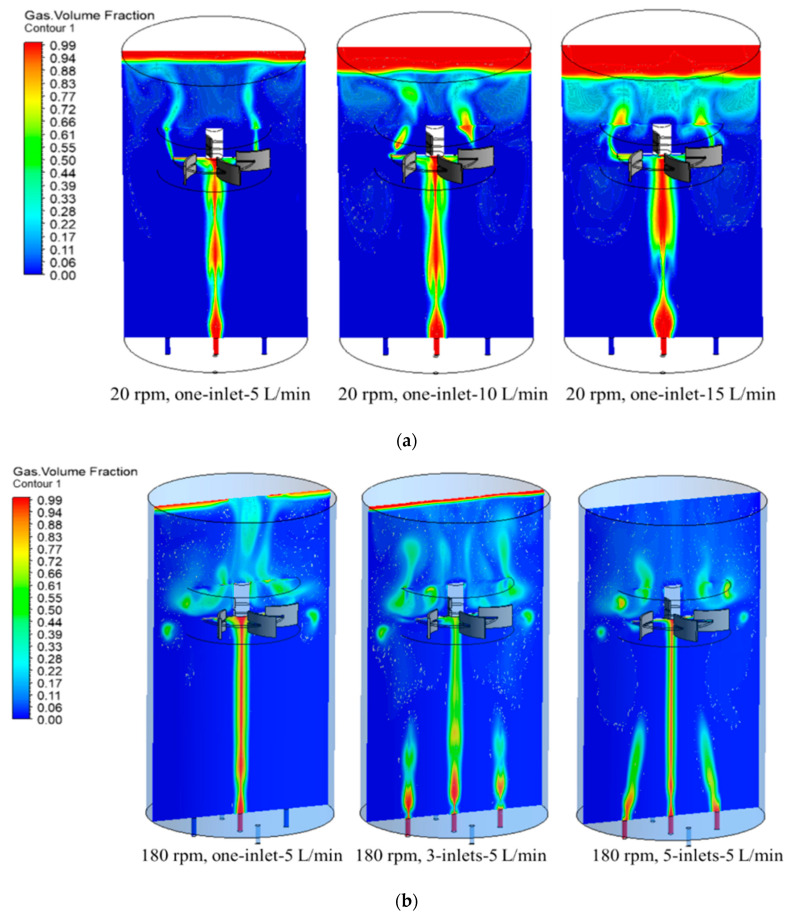
Effect of the spargers distribution on flow field in different speed and gas flow rates. (**a**) 0.62% gas volume fraction distribution at different flow rates when operating at 20 rpm, one sparger. (**b**) 0.62% gas volume fraction distribution with one, three, and five inlets at the 180 rpm and 5 L/min cases.

**Figure 9 entropy-24-00357-f009:**
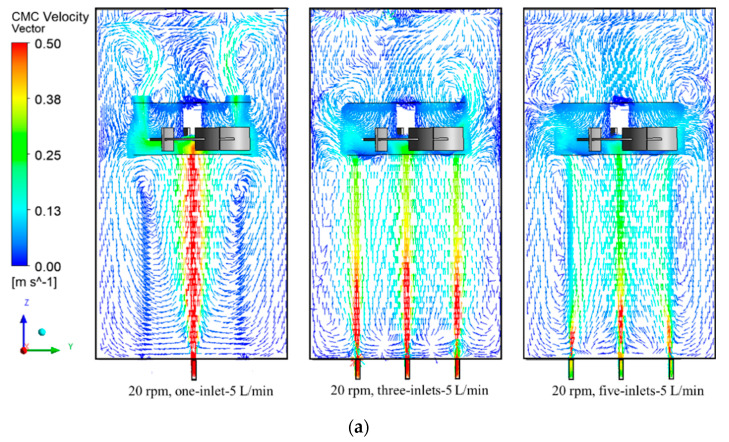
Effect of the spargers distribution on flow field in different speed and gas flow rates. (**a**) 0.62% CMC flow field when operating at 20 rpm and 5 L/min cases. (**b**) 0.62% CMC flow field when operating at 180 rpm and 10 L/min cases.

**Figure 10 entropy-24-00357-f010:**
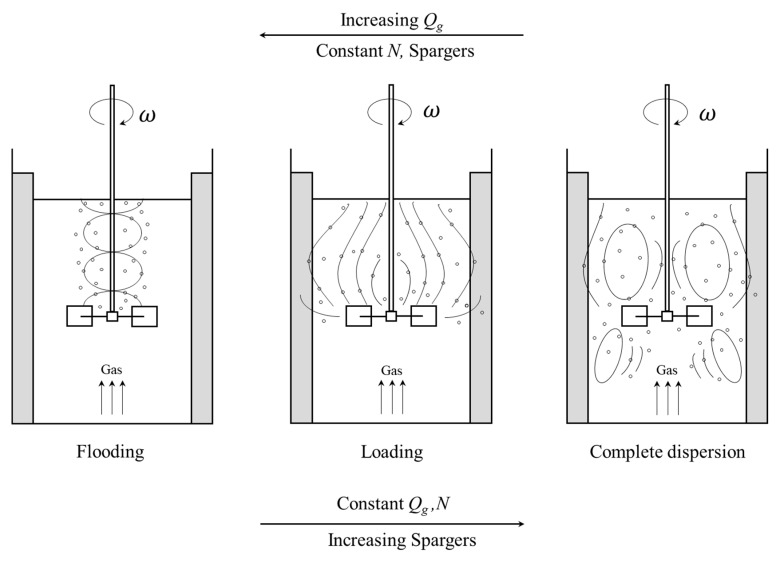
Flow regimes in a gassed vessel stirred by a radial flow impeller.

**Figure 11 entropy-24-00357-f011:**
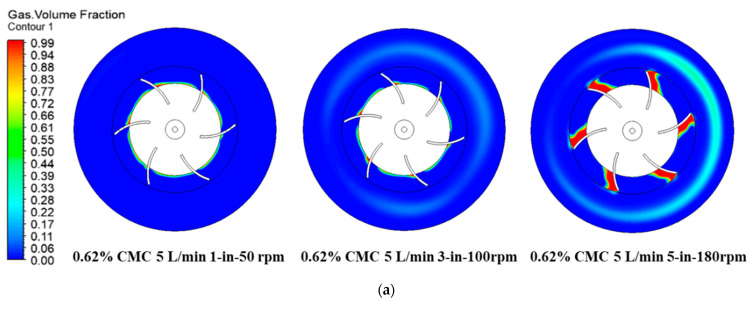
Effect of rheological properties, spargers, and rotation speeds on cavity formation. (**a**) Gas volume fraction distribution in 0.62%, 5 L/min with the increase of the rotation speeds and spargers. (**b**) Gas volume fraction distribution in 0.85%, 5 L/min with the increase of the rotation speeds and spargers. (**c**) Gas volume fraction distribution in 1.25%, 5 L/min with the increase of the rotation speeds.

**Table 1 entropy-24-00357-t001:** Geometrical dimensions of curved-blade Rushton turbine.

	Curved-Blade Rushton Turbine (CRT)			
	Impeller Parameters		Vessel Parameters	
$D_{s}$	[m]	0.114	$V_{T}$ [m^3^]	0.011
$W_{s}$	[m]	0.023	T [m]	0.190
$t_{S}$	[m]	0.002	$D_{s}/T$ [-]	0.600
$R_{S}$	[m]	0.043	H/T	2.000
$\alpha_{S}$	[°]	49.3	C/H	0.550

$D_{s}$ means the diameter of the impeller, $W_{s}$ means the heights of the impeller, $t_{S}$ means the blades thickness,$\;R_{S}$ means the radii of the blades, $\alpha_{S}$ means the radian angle, $V_{T}$ means the volume of the tank, T is diameter of the tank, *H* is the height of the tank, *C* is the clearance of the impeller.

**Table 2 entropy-24-00357-t002:** Rheological properties of the CMC solutions.

	C_1,2,3_ [Ma-%]	K_1,2,3_ [Pa·s^m^]	m_1,2,3_ [-]	$\eta_{0,1,2,3}\;[\mathrm{Pas}]$	${\eta_{\infty }}_{1,2,3}\;[\mathrm{Pas}]$	ρ_1,2,3_ [kg·m^3^]
Rheo1	0.62	1.75	0.464	0.02	0.79	1004
Rheo2	0.85	4.99	0.388	0.03	2.87	1005
Rheo3	1.25	15.34	0.325	0.14	11.98	1010

## Data Availability

Not applicable.

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
