# Peer review of "The Effect of the Spargers Design on the Wastewater Treatment of Gas-Liquid Dispersion Process in a Stirred Tank"

_entropy, 2022, doi:10.3390/e24030357_

Round 1

Reviewer 1 Report

See the attached pdf

Author Response

Dear Editor,

Based on the suggestion from the reviewers, we have carefully discussed the valuable comments from the reviewers and tried our best to revise the manuscript. A revised manuscript, with the correction sections red marked, was attached to the supplemental material for easy check/editing purposes. If you have any questions, please feel free to contact us.

The point to the point response to the reviewer’s comments is listed as follows:

Reviewer1#

  • Line 40: 40 define T

R: Thank you for your comments, where T is the tank diameter, is already added in the manuscript.

  • Line 41: also provide the main conclusions of Joshi et al

R: Thank you for your comments, Joshi et al. [10], reviewed the work on gas flow rate, impeller diameters mechanically stirred gas-liquid tank reactors, they found that the drag experienced by the bubble depends on both the Reynolds and Eotvos numbers.

  • Line 40: define ‘flooding case’, why is it relevant?

R: Thank you for your comments, the flooding has already been explained as “(flooding is one of the phenomena in gas-liquid mixing when high gas loadings the impeller no longer pumps the gas and liquid mixture adequately and the gas rises axially as a bubbling stream to the liquid surface).” in the manuscript.

  • Line 40: The effects of spargers design and location with a concave impeller on flow patterns

and power consumption in non-Newtonian fluids was investigated [sic]’ → explain here how this is done and how the forthcoming methodology is treated in the remainder of the paper.

R: Thank you for your comment, we presented the effect of those parameters on flow patterns and power consumption in the whole manuscript.

  • Line 91: the diameter T is given but the tank has more characterizing parameters (height, the position of the impeller, etc.); I suggest summarizing all these properties/dimensions in a table.

R: The details of the system geometry parameters were already added in Table 1.

6) Line 110: explain how this formula is used, both K and m depend on Ë™γ but it is not stated how this is accounted for in Eq. (1); provide a figure showing the behavior of µa versus Ë™γ.

R: Thank you for your comment, the figure showing how to obtain the K and m is already present as seen in Fig.2.

7) Line 123: here D is used to indicate the diameter (T was used earlier), or do you mean the

diameter of something else? please explain/correct

R: Thank you for your comment, here the D is the diameter of the inlet distribution located, but T is the diameter of the tank.

8) Line 127: the word ‘power consumption is used frequently and should be defined clearly: according to Eq.(2) it should be the net power of the impeller delivered to the fluid, so state this in the definition; the same remark holds for other key properties that should be introduced and defined precisely

R: Thank you for your comment, the power consumption is already explained in the introduction part, since it is commonly used in the mixing system, we did not define it.

9) Line 130: P is the net power delivery without gas, and P0 with gas; shouldn’t this be the opposite?

R: Thank you for your comment, you are correct, P is the net power delivery without gas, and P0 with gas.

10) Line 134: the numbers Ne and Redo not affect the mixing, they characterize it.

R: Thank you for your comment, you are correct, those two numbers are characteristic of the mixing phenomenon

11) Line 138: explain how the viscosity µeff relates to µa defined earlier in Eq.(1)

 R: Thank you for your comment, actually both are the same thing, just different calling, we already unified it in the manuscript.

12) 138:Eq.(1) is never used in the remainder of the paper, so why is it given R: Thank you for your comment, Eq.(1) is used in reference 33, also presented in Eq.(1), and also present how to obtained the power-law fluids fitted parameters in this reference.

13) 138:yet another definition of the diameter D (or does this refer to the diameter of something else?), please use consistent definitions throughout

R: Thank you for your comment, the diameter is the impeller diameter, we already present a table in our manuscript to explain those parameters in detail.

14) 161: formally, the nabla operator has an inner product: ∇ · (αiρiνi)

R: Thank you for your comment, we have already added the operator to the manuscript.

15) 163: the second term should read ∇·(αiρiνi⊗νi); correct the last two terms, they are incorrect and contradictory; explain the meaning of the symbols used in the accompanying text.

R: Thank you for your comment, we have already corrected the equation.

16) 164: I would use subscript i to indicate constituent i, αi the becomes its volume fraction,ρi its density (constant) and νi its velocity; Eq.(7) would then read P i=1,2 αi = 1.

 R: Thank you for your comment, we have already corrected the equation.

17) 168-171: Eqs.7-11: not clear how the k −ε equations are implemented and how the subscript k is used here; do k and ε relate to the liquid only, or both respective phases, or the mixture as a whole? please explain; what for instance means ρk? (the density

of k?)

R: Thank you for your comment, here we applied the mixture as a whole, we have already corrected the equation.

18) 177: where does the drag force FD appear in the momentum equation?

R: Thank you for your comment, the contributions to the forces acting on the dispersed phase, the volume-averaged momentum equations are formulated based on the equation of a single fluid particle motion. After that, one-dimensional momentum equations are newly obtained from the averaged equations. It is shown that the wall drag term in the dispersed phase is associated with the spatial gradient of the volume-averaged viscous stress of the continuous phase. The magnitude of the wall drag term for a phase is its volume fraction multiplied by the total two-phase pressure drop induced by the wall shear of the continuous phase.

19) 188: it is not made clear how this equation is used to calculate the various parameters in the formulation of the drag force, moreover, the symbols used in the equation (e.g. the bubble velocity ub) are inconsistent with those used in the earlier Eqs.5-6

R: Thank you for your comment, we have already checked the equation, the ub is the local velocity of bubbles, and the source term includes bubble sink or source in different situations including coalescence, breakage, phase interactions, reaction, and mass transfer. The Lue and Svendsen models have been utilized in this study suggested by ANSYS Fluent user guide to model the bubble coalescence and breakup.

20) 197: define QG

R: Thank you for your comment, the QG is the gas flow rate, just a demonstration of the change of flow rate and speed effect on the gas-liquid flow.

21) 201: I am surprised that no-slip boundary conditions are used on the tank wall since this requires a very fine mesh (locally) to resolve the laminar sublayer; maybe this is possible because the Reynolds- numbers are comparatively low, anyway, explain a bit more here, also using Eq.17 (which should be used in the discussion on the mesh generation and b.c.’s, it has nothing to do with the time step size); also mention here the boundary conditions used at the impeller as this will significantly influence the flow at the impeller

R: Thank you for your comment, since our simulation is with the lower speed (lower Re), combined with our very fine mesh as presented in Fig.4, the near-wall region is already refined during the mesh production process.

22) 201: I suggest splitting Section 3 into two sections (3 and 4), the first dealing with the experiments and model validation, the second treating the parameter study that is performed with the numerical model

R: Thank you for your suggestion, we would like to present the result as the author’s writing style.

23) 214: Section 3.1: the agreement between model and experiment is very good, considering Re vs. Ne; however, the experimental results deserve a more general discussion (are the flow patterns and gas distribution reproduced well?) which will then also identify the important mechanisms to be studied later on in the parameter study

R: Thank you for your question, the flow patterns and gas distributions results from the experiment will be conducted in our next research work.

24) 236: it is not explicitly stated that Sections 3.3 - 3.4 are based on the numerical model runs (are they...?), while it is also confusing that Section 3.2 of the parameter study uses the experiments

R: Thank you for your comment, we presented the results firstly to validated the simulation results, and then present the experiment results in section 3.2, from section 3.3, we mainly present the CFD results.

25) fig 5 the purpose of this Section (influence of sparger configuration on power draw) is better illustrated when combining the results for the same gas flow is one figure

(i.e. plot simultaneously 5 L/s for 1, 3, 5 inlets in one figure)

R: Thank you for your comment, in this part, we are mainly concerned about the effect of sparger distribution on the power consumption, we just pick up some of our results to discuss.

26) 302: is fig 6 a numerical or experimental result? indicate this in the caption; the same

the remark applies to all remaining figures

R: Thank you for your comment, Fig.6 is the simulation results, we already indicated in the manuscript.

27) 359: cavity formation is influenced by the boundary conditions imposed on the impeller blades, which should be discussed in this section; are the results consistent for different choices of boundary conditions (slip, no-slip, partial slip)?

R: Thank you for your comment, in this manuscript, all cases are handled as the no-slip boundary condition.

Reviewer 2 Report

The subject of this paper is the application of the CFD-PBM simulation method by means of a commercial software tool to study the gas-liquid dispersion in a stirred tank reactor. Here, the continuous phase is a non-Newtonian medium in form of CMC solution, which is used as a model fluid for wastewater. The stirring geometry is a novel curved-blade Rushton turbine. Here, the validation with experimental data is performed for three different solutions with different CMC concentrations for different gas volume flows and different numbers of inlets. For validation purposes, the Newton number (Power number) is considered as a function of the Reynolds number. Detailed experimental studies on the influence of the number of gas distributors for different CMC concentrations and gas volume flows follow. From the numerical simulations, the gas distributions and flow fields for different stirrer speeds and gas volume flows are also presented. In the last section, the formation of cavities at three different CMC concentrations and three different stirrer speeds each is investigated by means of numerical simulation. The results are relevant to practice. However, I have some comments on the manuscript.

Line 40: It should already be explained here what T is.

Line 82: At this point, the term "flooding" should be explained for the first time.

Please always use a space between the number and the unit and a space between the word and the bracket, for example, “population balance model (PBM)”

Chapter 2.1: A graphical plot of the measured values (apparent viscosity as a function of shear rates) to determine K and m would be helpful.

Line 123: Please refer to Figure 2 when mentioning the diameter D.

Equation 4: In the further text, the Newton number is always referred to the gas-liquid stirred tank. Therefore, please make clear in equation 4 that Ne_G is used. In line 134 a wrong variable was used for the Ne-number (N_e instead of Ne). Please also mention that the Newton number is the Power number.

Lines 149/150: Which target variable was considered in the grid independence study? Please provide further details on the grid independence study.

Lines 173/174: Where do the parameters are given here (C_eta, C_1, C_2 etc.) come from?

Lines 191/192: Please explain the model of Lue and Sevendsen in more detail. Line 192: “guide” instead of “guider”

Chapter 2.2.2: Please provide the complete balance of forces. Has the Basset force also been taken into account in equation 6?

Why are bubble size distributions not considered in this research? I miss diagrams showing the simulated bubble size distribution.

Is there data to validate the gas volume fraction distribution, especially at low gas contents (determined for example by optical methods, the stirred reactor is after all transparent).

Figure 5 f): The caption is missing.

Figure 6 a) Please note that 0.62% is the CMC concentration. Is it by weight percent?

Figure 6 b): The caption is wrong. It is about the gas distribution and not about the flow field.

Figure 7: Please note that this is the absolute velocity. Again, is there any experimental data (e.g. Particle Image Velocimetry data) of the flow field that can be used for validation?

Figure 8: This diagram is very helpful. However, the text also considers the impeller speed as a changing parameter. Can the principle sketch still be extended if N is not constant (representation of the influence of N)?

Figure 9: They are very nice results that are well explained. At this point, the influence of the shear rate and the size of the viscosity in the immediate vicinity of the stirrer blades would also be very interesting.

Can you say something about the scale-up? What would happen with larger stirred tanks (larger volume)? Can the simulation method used here also be transferred to the fully turbulent range? For biotechnological applications, these speeds are sufficient in the laminar range. Is transferability also possible for higher speeds?

Also, please check all the text regarding the English language. For example, I suggest the following sentence part for lines 79/80: “Rielly and Nagy [32] utilized the CFD to study the gas-liquid stirred system and pointed out that the concave shape of the RT impeller produced a smaller gas cavity behind the blades.” Or lines 147/148: “The structured mesh of the whole impeller after mesh rotation of the mesh in Fig.3.” (incomplete sentence, the verb is missing). Or lines 383/384: “Compared with one sparger, the increasing spargers helps the gas bubbles distribute more even in the investigated plane.”

Reference list: For some references, the title of the publication is given in quotation marks, for some it is not. Please check. Reference 33: The year is missing.

Author Response

Dear Editor,

Based on the suggestion from the reviewers, we have carefully discussed the valuable comments from the reviewers and tried our best to revise the manuscript. A revised manuscript, with the correction sections red marked, was attached to the supplemental material for easy check/editing purposes. If you have any questions, please feel free to contact us.

The point to the point response to the reviewer’s comments is listed as follows:

Reviewer #2:

1)Line 40: It should already be explained here what T is.

R: Thank you for your comments, where T is the tank diameter, is already added in the manuscript.

2)Line 82: At this point, the term "flooding" should be explained for the first time.

R: Thank you for your comments, the flooding has already been explained as “(flooding is one of the phenomena in gas-liquid mixing when high gas loadings the impeller no longer pumps the gas and liquid mixture adequately and the gas rises axially as a bubbling stream to the liquid surface).” in the manuscript.

3)Please always use a space between the number and the unit and a space between the word and the bracket, for example, “population balance model (PBM)”.

R: Thank you for your comments, all the needed spaces were already added to the manuscript.

4)Chapter 2.1: A graphical plot of the measured values (apparent viscosity as a function of shear rates) to determine K and m would be helpful.

R: Thank you for your comments, the graphical plot was already added to the manuscript.

5)Line 123: Please refer to Figure 2 when mentioning the diameter D.

R: Thank you for your comments, the manuscript has already been modified as “The 4 orifices evenly distributed on the circumference a diameter of D/3 (as seen in Fig.2).”

6)Equation 4: In the further text, the Newton number is always referred to as the gas-liquid stirred tank. Therefore, please make clear in equation 4 that Ne_G is used. In line 134 a wrong variable was used for the Ne-number (N_e instead of Ne). Please also mention that the Newton number is the Power number.

R: Thank you for your comments, the mentioned already modified in the manuscript.

7)Lines 149/150: Which target variable was considered in the grid independence study? Please provide further details on the grid independence study.

R: Thank you for your comments, the power number was chosen as the target variable and the independence analysis as shown in Fig.5.

8)Lines 173/174: Where do the parameters are given here (C_eta, C_1, C_2, etc.) come from?

R: Thank you for your comments, those data have come from the Analysis Fluent user guide.

9)Lines 191/192: Please explain the model of Lue and Svendsen in more detail. Line 192: “guide” instead of “guider”

R: Thank you for your comments, the Lue and Svendsen models were suggested by ANSYS Fluent user guide, which we just implanted here.

10)Chapter 2.2.2: Please provide the complete balance of forces. Has the Basset force also been taken into account in equation 6?

R: Thank you for your comments, the Lue and Svendsen models were suggested by ANSYS Fluent user guide, which we just implanted here.

11)Chapter 2.2.2: Please provide the complete balance of forces. Has the Basset force also been taken into account in equation 6?

R: Thank you for your suggestion, the complete balance of forces has already been added. The viscous was already taken into account in the equation.

12)Why are bubble size distributions not considered in this research? I miss diagrams showing the simulated bubble size distribution.

R: Thank you for your comments, we have taken the bubble size into account in this research.

13)Is their data to validate the gas volume fraction distribution, especially at low gas contents (determined for example by optical methods, the stirred reactor is after all transparent).

R: Thank you for your comments, this is our next step in research work.

14)Figure 5 f): The caption is missing.

R: Thank you for your comments, the missing caption has already been added.

15)Figure 6 a) Please note that 0.62% is the CMC concentration. Is it by weight percent?

R: Thank you for your comments, the concentration of CMC solution was prepared by mass content as described in section 2.1.

16)Figure 6 b): The caption is wrong. It is about the gas distribution and not about the flow field.

R: Thank you for your comments, the wrong caption was already modified in the manuscript.

17)Figure 7: Please note that this is the absolute velocity. Again, is there any experimental data (e.g. Particle Image Velocimetry data) of the flow field that can be used for validation?

R: Thank you for your comments, the PIV data is our next step in research work.

18)Figure 8: This diagram is very helpful. However, the text also considers the impeller speed as a changing parameter. Can the principle sketch still be extended if N is not constant (representation of the influence of N)?

R: Thank you for your comments, as the rotation speed N is not our research interest.

19)Figure 9: They are very nice results that are well explained. At this point, the influence of the shear rate and the size of the viscosity near the stirrer blades would also be very interesting.

R: Thank you for your comments, the effect of rheological properties, inlet distributions, and rotation speeds on cavity formation was already extensively discussed here.

20)Can you say something about the scale-up? What would happen with larger stirred tanks (the larger volume)? Can the simulation method used here also be transferred to the fully turbulent range? For biotechnological applications, these speeds are sufficient in the laminar range. Is transferability also possible for higher speeds?

R: Thank you for your comments, scaling up of mixing operation from a laboratory or pilot plant requires that the physical and chemical properties of the product are similar at the full-scale plant level. It also requires that the desired outcome is produced within a reasonable amount of time. Below are some items worth considering before and during the scale-up phase of any mixing in-tank operation. The geometric similarity is often used in mixing scale-up because it greatly simplifies design calculations. Geometric similarity means that a single ratio between small scale and large scale applies to every length dimension. With geometric similarity, all of the length dimensions in the large-scale equipment are set by the corresponding dimensions in the small-scale equipment. Besides that also other parameters must be remained, like the geometrical shape of the bottom, impeller emplacement, and number and baffles. The simulation method can be transferred to the fully turbulent range. It is possible for operating at higher speeds.

21)Also, please check all the text regarding the English language. For example, I suggest the following sentence part for lines 79/80: “Rielly and Nagy [32] utilized the CFD to study the gas-liquid stirred system and pointed out that the concave shape of the RT impeller produced a smaller gas cavity behind the blades.” Or lines 147/148: “The structured mesh of the whole impeller after mesh rotation of the mesh in Fig.3.” (incomplete sentence, the verb is missing). Or lines 383/384: “Compared with one sparger, the increasing spargers helps the gas bubbles distribute more even in the investigated plane.”

R: Thank you for your comment, we already found a native speaker to help us to polish the language.

22)Reference list: For some references, the title of the publication is given in quotation marks, for some it is not. Please check. Reference 33: The year is missing.

R: Thank you for your comments, we have already double-checked the reference.

Round 2

Reviewer 1 Report

Authors have addressed most of the minor comments I had on the first draft manuscript. However, the major issues concerning the description of the research method and the presentation and validation of the results have not been improved (see previous review report: General comments). Some equations still contain errors (forcing term in the momentum equation) or use undefined symbols (k-epsilon equation). It is not explained how the bubble density n is used certain quantities in the flow model. The addition of Figure 1 and Table 1 adds little information since important values/descriptions are missing.   

Author Response

  1. Authors have addressed most of the minor comments I had on the first draft manuscript. However, the major issues concerning the description of the research method and the presentation and validation of the results have not been improved (see previous review report: General comments). Firstly, the authors should be more explicit, from the beginning, in explaining the overall methodology of their work. It should be made more clear how the experiments and CFD model are related and used to derive conclusions later on. In this respect, it appears to me that Paragraphs 3.3 - 3.4 are entirely based on CFD results, while Section 3.2 is not, but this is never mentioned by the authors.

R2: Thank you for your comment, we have addressed the overall structure or layout of this manuscript in the end of the introduction section as following: “ The remaining part of the paper is organized as follows: Section 2 provides a detailed description of the experimental set-up and methods for measuring the rheology of non-Newtonian fluids, obtained the torque when operating in different rotation speed characterization, the CFD methods and basic theory used in this paper also been presented; Section 3.1 firstly provides a brief comparison between the measured torque and the CFD predict value for validating the proposed CFD method, in Section 3.2 provides the measured power in different operation conditions and from section 3.3 to end presents the CFD simulation results. The salient conclusions are highlighted in Section 4.”

  1. -Some equations still contain errors (forcing term in the momentum equation) or use undefined symbols (k-epsilon equation).

-All mathematical formulas, especially those in Section 2, should be reviewed for correctness and consistency; do not forget to explain the meaning of all variables and symbols in the accompanying text!

R2: Thank you for your suggestion, we have exam all related equations in section 2 and explain the meaning of all variables and symbols in the accompanying text.

  1. It is not explained how the bubble density n is used certain quantities in the flow model. The addition of Figure 1 and Table 1 adds little information since important values/descriptions are missing.

R2: Thank you for your suggestion, we have already explained how the bubble density is used certain quantities in the flow model. we described as:

“The PBM method describes the bubble in/out defined control volume.

      (17)

where presents bubble density distribution function at the time of t and a position of is the volume of bubbles, is the local velocity of bubbles, and is the source term including bubble sink or source in different situations including coalescence, breakage, phase interactions, reaction, and mass transfer. The Lue and Sevendsen models have been utilized in this study suggested by ANSYS Fluent user guide to model the bubble coalescence and breakup.”

  1. The addition of Figure 1 and Table 1 adds little information since important values/descriptions are missing.

R2: Thank you for your suggestion, to increase the readability, we add the detailed regression data from the CMC solution as shown in Table 2.

Table 2. Rheological properties of the CMC solutions

C [Ma-%]

K [Pa.sm]

m [-]

 [Pas]

 [Pas]

ρ [kg.m3]

Rheo1

0.62

1.75

0.464

0.02

0.79

1004

Rheo2

0.85

4.99

0.388

0.03

2.87

1005

Rheo3

1.25

15.34

0.325

0.14

11.98

1010

Reviewer 2 Report

Based on the detailed responses and corrections, the manuscript can be accepted.

Author Response

Thank you so much for your the time to review our paper.

Round 3

Reviewer 1 Report

The following improvements must be made:

  1. the accompanying text to Eq. 1 should also mention that the three values for the coefficients K and m, respectively, concerns three different CMC solutions. Also, state this clearly in the caption of Table 2.
  2. what information is Fig. 2 supposed to convey? Is of little use without further explanation.
  3. Table 1: mention the meaning of the different symbols that are used.
  4. I am still confused about the definitions of the power P and P0 (page 7), I would expect P0 to be the power consumption of the unaerated case, but apparently this is not so?
  5. Eq. 5 and onwards: ALL symbols used in the equations should be clarified in the accompanying text (not only alpha_i and v_i); how if the force F_j specified? Why do you use a subscript j here, instead of i as for the other quantities; this may seem a trifle but someone not familiar with your work has a hard time understanding these equations.
  6. Eq. 8-9: if alpha_m is the volume fraction of the mixture, it should be equal to one, right? (so why use it as an additional parameter?)
  7. despite the comment by the authors, it is still not clear to me how the bubble density is used in the equations. => "The PBM method describes the bubble in/out defined control volume.", I know quite a bit about numerical modelling, but do not know what is meant here. It could be that the bubble density is an interesting output parameter in itself, but the authors suggest it is used in the flow model as well. Please clarify!
  8. Eq. 15: how is the Reynolds-number defined here? I suppose it is different from the definition in Eq. 3. Overall, you should give sufficient information to someone who might want do you some simulations him/herself.
  9. I suggest to change the titles of subparagraphs 3.1-3.4 as follows (or similar) to improve readability: (3.1): Experimental validation of CFD model. (3.2): Experimental observations of the effect of sparger distribution on power consumption. (3.3) Numerical assessment of the effect of sparger distribution on mixing flow patterns. (3.4) Numerical assessment of the effect of sparger distribution on cavity formation.   

Author Response

Dear Editor,

Based on the suggestion from the reviewers, we have already discussed the valuable comments from the reviewer and tried to response the reviewer’s comments and revise the manuscript.

The point to the point response to the reviewer’s comments is listed as follows:

Reviewer1#

  1. the accompanying text to Eq. 1 should also mention that the three values for the coefficients K and m, respectively, concerns three different CMC solutions. Also, state this clearly in the caption of Table 2.

R:Thank you for your comment, we already identified K and m concerns three different CMC solutions, see: K1 = 1.75, K2 = 14.99, K3 = 115.34 Pa·sm, m1 = 0.464, m2 = 0.388, m3 = 0.325, which yielding R21,2,3=0.994,0.996,0.995, respectively,

  1. what information is Fig. 2 supposed to convey? Is of little use without further explanation.

R: Thank you for your comment, the figure 2 is just a demonstration of how we obtained the K and m from the raw data, the regression process which obtained the K and m as seen in the Table 2 and Fig.2.

  1. Table 1: mention the meaning of the different symbols that are used.

R: Thank you for your comment, the meaning of different symbols now been added in the manuscript,  means the diameter of the impeller,  means the heights of the impeller,  means the blades thickness, means the radii of the blades,  means the radian angle, means the volume of the tank,  is diameter of the tank, H is the height of the tank, C is the clearance of the impeller.  

  1. I am still confused about the definitions of the power P and P0 (page 7), I would expect P0 to be the power consumption of the unaerated case, but apparently this is not so?

R: Thank you for your comment, actually, the P is the power consumption of the unareated case, and the P0 is the aerated case.

  1. Eq. 5 and onwards: ALL symbols used in the equations should be clarified in the accompanying text (not only alpha_i and v_i); how if the force F_j specified? Why do you use a subscript j here, instead of i as for the other quantities; this may seem a trifle but someone not familiar with your work has a hard time understanding these equations.

R: Thank you for your comment, the equations all clarified in the accompanying text, the force is shows the momentum transferred from bubbles to the liquid phase and the last term in the right side of Eq.(6) is the inter-phase forces, and the subscript j already removed.

  1. Eq. 8-9: if alpha_m is the volume fraction of the mixture, it should be equal to one, right? (so why use it as an additional parameter?)

R: Thank you for your valuable comment, it should be equal to one, the alpha_m was removed from the equation.

  1. despite the comment by the authors, it is still not clear to me how the bubble density is used in the equations. => "The PBM method describes the bubble in/out defined control volume.", I know quite a bit about numerical modelling, but do not know what is meant here. It could be that the bubble density is an interesting output parameter in itself, but the authors suggest it is used in the flow model as well. Please clarify!

R: Thank you for your comment, we already modified the balance equation as follow:

      (17)

where the balance equation in terms of volume fraction of particle ‘i’ was expressed,is the density of secondary phase ‘p’ (kgm3),  is the volume fraction of particle‘i’(%),  is the volume of the particle‘i’(m3),  is the particle birth rate due to aggregation (m-3s-1),  is the particle death rate due to aggregation (m-3s-1),  is the particle birth rate due to breakage (m-3s-1), is the particle death rate due to breakage (m-3s-1). The growth rate of particle‘i’, which is (m3s1) of volume‘Vi’is defined as:

                                        (18)

The volume fraction of particle ‘i’, αi (%) was defined as Eq. (18), where,  is the local average number density of particle ‘i’ (m−3s-1) and was expressed as Eq. (19), where,  is the number density function.

                                     (19)

                                (20)

  1. Eq. 15: how is the Reynolds-number defined here? I suppose it is different from the definition in Eq. 3. Overall, you should give sufficient information to someone who might want do you some simulations him/herself.

R: Thank you for your comment, the Re number used in in the equation is changed as Reb which is different from the Re defined as in Eq. (3). Here Re = in Newtonian fluids.

  1. I suggest to change the titles of subparagraphs 3.1-3.4 as follows (or similar) to improve readability: (3.1): Experimental validation of CFD model. (3.2): Experimental observations of the effect of sparger distribution on power consumption. (3.3) Numerical assessment of the effect of sparger distribution on mixing flow patterns. (3.4) Numerical assessment of the effect of sparger distribution on cavity formation.   

R: Thank you for your suggestion, we accept the subparagraphs changes in the manuscript.
